# Assessment of Quercetin Antiemetic Properties: In Vivo and In Silico Investigations on Receptor Binding Affinity and Synergistic Effects

**DOI:** 10.3390/plants12244189

**Published:** 2023-12-18

**Authors:** Raihan Chowdhury, Md. Shimul Bhuia, Asraful Islam Rakib, Rubel Hasan, Henrique Douglas Melo Coutinho, Isaac Moura Araújo, Irwin Rose Alencar de Menezes, Muhammad Torequl Islam

**Affiliations:** 1Department of Pharmacy, Bangabandhu Sheikh Mujibur Rahman Science and Technology University, Gopalganj 8100, Bangladesh; raihanpharmacy049@gmail.com (R.C.); shimulbhuia.pharm@gmail.com (M.S.B.); asrafulislam.rakib.phr@gmail.com (A.I.R.); rubelhasanphr2@gmail.com (R.H.); 2Department of Biological Chemistry, Regional University of Cariri—URCA, Crato 63105-000, Brazil; hdmcoutinho@gmail.com (H.D.M.C.); isaac.moura@urca.br (I.M.A.)

**Keywords:** antiemetic, D2 receptor, molecular docking, Quercetin, toxicity

## Abstract

Quercetin (QUA), a flavonoid compound, is ubiquitously found in plants and has demonstrated a diverse range of biological activities. The primary objective of the current study is to assess the potential antiemetic properties of QUA using an in vivo and in silico approach. In this experiment, 4-day-old chicks were purchased to induce emesis by orally administering copper sulfate pentahydrate (CuSO_4_·5H_2_O) at a dose of 50 mg/kg (orally). Domperidone (DOM) (6 mg/kg), Hyoscine (HYS) (21 mg/kg), and Ondansetron (OND) (5 mg/kg) were treated as positive controls (PCs), and distilled water and a trace amount of Tween 80 mixture was employed as a negative control (NC). QUA was given orally at two distinct doses (25 and 50 mg/kg). Additionally, QUA (50 mg/kg) and PCs were administered separately or in combination to assess their antagonistic or synergistic effects on the chicks. The binding affinity of QUA and referral ligands towards the serotonin receptor (5HT3), dopamine receptors (D2 and D3), and muscarinic acetylcholine receptors (M1–M5) were estimated, and ligand–receptor interactions were visualized through various computational tools. In vivo findings indicate that QUA (25 and 50 mg/kg) has a significant effect on reducing the number of retches (16.50 ± 4.65 and 10.00 ± 4.19 times) and increasing the chick latency period (59.25 ± 4.75 and 94.25 ± 4.01 s), respectively. Additionally, QUA (50 mg/kg) in combination with Domperidone and Ondansetron exhibited superior antiemetic effects, reducing the number of retches and increasing the onset of emesis-inducing time. Furthermore, it is worth noting that QUA exhibited the strongest binding affinity against the D2 receptor with a value of −9.7 kcal/mol through the formation of hydrogen and hydrophobic bonds. In summary, the study found that QUA exhibited antiemetic activity in chicks, potentially by interacting with the D2 receptor pathway.

## 1. Introduction

Emesis, commonly referred to as vomiting, is a well-recognized physiological reflex that can be triggered by various peripheral and central stimuli [1]. These stimuli encompass gastrointestinal irritation or inflammation, the presence of cancer chemotherapy, circulating toxins and other pharmacological agents, pain, anxiety, and motion that provokes a response [2]. The defensive reflex is a commonly observed phenomenon in both animals and humans. In contrast, nausea is a subjective experience characterized by a disagreeable sensation closely linked to the act of vomiting [3].

The pathophysiology of vomiting is characterized by a significant degree of complexity. The reticular formation encompasses a vomiting center (VC), which can be stimulated by afferent stimuli originating from the gastrointestinal tract (GIT) or by the chemoreceptor trigger zone (CTZ) [4,5].The CTZ plays a regulatory role in various physiological processes, including the control of food intake, conditioned taste aversion, and GIT motility [6,7]. The initiation of emesis occurs when the CTZ detects emetogenic toxins in the bloodstream and cerebrospinal fluid (CSF) [8]. This detection is facilitated by various receptors within the CTZ, such as the serotonin receptor (5HT3), muscarinic acetylcholine receptors (M1-M4), dopamine receptors (D2, D3), histamine (H1), opioid receptors, and neurokinin 1 receptor (NK1) for substance P [9,10,11]. Once these receptors detect the emetogenic toxins, they transmit a signal to the nearby nucleus tractus solitaries (NTS) [12,13]. The NTS functions as the principal mechanism by which all emetic stimuli elicit the physiological response of vomiting, providing a comprehensive final pathway. During the act of emesis, the abdominal muscles undergo a retrograde contraction while the lower esophageal sphincter relaxes, facilitating the expulsion of stomach contents through the oral cavity and inducing retching [14,15].

The D2 receptor has a role in several cognitive and behavioral processes, including movement, attention, sleep, memory, and learning [16]. In addition, the D2 receptor plays a vital role in the physiology of eliciting emesis [17]. The precise signaling processes behind the induction of vomiting by the D2 receptor have yet to be elucidated. The research evidence suggests that the activation of extracellular signal-regulated kinase (ERK)-, phosphatidylinositol-3 kinase (PI3K)-, and protein kinase C (PKC)-related signaling cascades via the D2 receptor might possibly lead to the occurrence of emesis [18,19].

At present, a diverse range of antiemetic medications that are approved for clinical use are utilized in the treatment and control of symptoms associated with nausea and vomiting. The medications can be classified into various categories, for instance, anti-dopaminergic drugs, antihistamines, serotonin antagonists, NK1-receptor inhibitors, anticholinergic drugs, and receptor agonists for corticosteroids, gamma-aminobutyric acid (GABA) type B receptor, and cannabinoid receptor type 1 (CB1) [20,21]. The extended utilization of synthetic antiemetic medications is also associated with various adverse effects, such as muscle weakness, spasms, or convulsions [22,23]. Hence, natural products have become a crucial requirement in contemporary society owing to their minimal adverse effects and economic advantages [24,25]. Current efforts in the exploration of new antiemetic drugs derived from natural sources are primarily centered around mechanism-based strategies that target specific cellular and molecular mechanisms. Polysaccharides, flavonoids, alkaloids, diterpenes, glucosides, cannabinoids, diarylheptanoids, chalcones, saponins, hydroxycinnamic acids, phenylpropanoids, terpenes, and lignans encompass a wide range of bioactive compounds that are of interest in the exploration of potential candidates for the development of antiemetic drugs [7,24,26].

The compound known as Quercetin (QUA), with the IUPAC (International Union of Pure and Applied Chemistry) name of 2-(3,4-dihydroxy phenyl)-3,5,7-trihydroxychromen-4-one, is a polyphenolic flavonoid that exhibits a broad distribution. A diverse array of fruits, vegetables, grains, and leaves, such as grapes, green tea, apples, citrus fruits, berries, cherries, capers, leafy greens, kale, and red onions, are known to contain compounds that exhibit a wide range of health-promoting effects in relation to various diseases [27,28,29]. QUA possess a wide range of applications in the production of nutritional supplements, beverages, and food products [30]. According to numerous studies, QUA has been found to possesses various pharmacological activities, including but not limited to anticancer [31], antioxidant [32], anti-obesity [33], anti-aging [34], anti-inflammatory [35], antidiabetic [36], antiallergic [37], cardioprotective [38], immune modulatory [39], neuroprotective [40], antiviral, and antimicrobial effects [41]. Additionally, QUA has a gastroprotective effect on experimental animals. QUA treatment significantly reduces the number of mast cells and impedes the site of gastric erosion [42]. Similarly, QUA exhibits a protective effect on stomach mucosal epithelial cells, safeguarding them from harm induced by ischemia-reperfusion, peroxide exposure, and bacterial infection [43,44,45]. In a study conducted by Lee et al., 2005, the researchers examined the impact of QUA on the activity of the mouse 5HT3 channel, a receptor known to be involved in the physiological process of vomiting [46].

There are various in vivo and in vitro models available for evaluating the antiemetic properties of a compound or plant extract. The chick emesis model is widely regarded as one of the available models for investigating emesis [20]. We conducted the QUA antiemetic test in the chick emesis model, which has not been investigated yet. In addition, computer-aided drug design (CADD) techniques have become more significant in the field of drug discovery [47]. Molecular docking is the prevailing computational approach that indicates the affinity of ligand–receptor interactions and facilitates the estimation of the specific location of contact [48]. The primary objective of this study is to assess the potential antiemetic effect of QUA in chicks (in vivo). Moreover, we also carried out an in silico study to identify potential molecular binding interactions with the emesis-inducing receptors underlying the observed effect.

## 2. Results

### 2.1. In Vivo Investigation

#### CuSO_4_·5H_2_O-Induced Emesis Test

In the in vivo experiment, it was observed that administering two distinct doses of QUA resulted in a significant dose-dependent elevation in the onset of retching in the experimental chickens, as compared to the NC group. The onset periods for the NC group were 8.25 ± 2.88 s. Simultaneously, the latency of the (Quercetin-25 mg/kg) QUA-25 and (Quercetin-50 mg/kg) QUA-50 groups were 59.25 ± 4.75 and 94.25 ± 4.01 s, respectively. On the contrary, in the different PC groups (DOM, OND, and HYS) administered, the animals demonstrated a higher latency period compared to the NC but less than that of the highest test dose of the group (QUA-50); the latency periods were 61.00 ± 3.34, 16.17 ± 3.64, and 11.50 ± 2.38 s, for the standard groups DOM, OND, and HYS, respectively. It is noteworthy that combining QUA with PC resulted in a higher latency period than administering PC or the different test doses alone. The highest latency period was noticed in the DOM + QUA-50 group, and the animal in this group showed delayed initiation of the retching by 156.67 ± 2.40 s. The onset of retching for the other two combined groups was noted at 146.33 ± 4.09 and 128.20 + 4.15 s for the OND + QUA-50 and HYS + QUA-50 groups, respectively. Figure 1 illustrates the latency observed across all treatment groups.

The NC group exhibited the highest number of retches, with a mean value of 64.25 ± 3.69. The frequency of retching in the experimental group decreased progressively as the dosage of the test sample (QUA) increased. For instance, QUA-25 and QUA-50 demonstrated 16.50 ± 4.65 and 10.00 ± 4.19 retches, respectively; this is a dose-dependent response. On the other hand, the animal given standard drugs also revealed a significant diminishing in the frequency of retching, and we observed a mean of retching of 11.80 ± 2.09, 30.80 ± 4.04, and 48.83 ± 4.17 for DOM, OND, and HYS, respectively. The administration of combination drug therapy remarkably decreased the frequency of retches compared to the administration of the drug alone, both the test compound and the standard. The DOM + QUA-50 manifested the lowest number of retches, with a mean value of 6.00 ± 2.52. The other two combination therapies also revealed a lower number of retches, with a mean value of 8.33 ± 1.20 and 12.2 ± 2.15 for OND + QUA-50 and HYS + QUA-50, respectively. Figure 2 displays the total count of retches observed across all treatment groups.

In a comparison of the results among the different treatment groups, we found that the highest percentage increase in latency was 94.73% for the DOM + QUA-50 group. A dose-dependent elevation of the increase in latency was observed in the test groups (QUA-25 and QUA-50). The fast onset of retching was noticed in the HYS group, and animals treated with HYS revealed a 28% increase in latency, indicating the most vital effect of CuSO_4_·5H_2_O among the different PC and test groups. As with latency, we also calculated the highest protection from the effect of retching in the DOM + QUA-50 group, and animals in this group revealed a 90.66% decrease in retches. The test sample demonstrated a dose-dependent percentage decrease in retching. QUA-25 and QUA-50 expressed 74.32% and 84.43% percentage decreases in retches, respectively. The percentage increase in latency and percentage decrease in retching of all the treatment groups are provided in Table 1.

### 2.2. In Silico Investigation

#### 2.2.1. QUA and DOM with Dopaminergic (D2 and D3) Receptor Interaction

Both QUA and standard DOM exhibited higher binding affinity toward the D2 receptor (Table 2). The binding affinities of QUA for the D2 and D3 receptors were −9.7 and −8.5 kcal/mol, respectively. The interaction between QUA and the D2 receptor involved the formation of four hydrogen bonds with specific amino acid residues, namely SER193 (1.85 Å), TYR416 (2.20 Å), THR119 (2.14 Å), and CYS118 (2.18 Å). Additionally, QUA formed three hydrophobic bonds with ASP114, TRP386, and PHE390 amino acids at the receptor binding site. In contrast, it is worth noting that DOM exhibited binding affinities with D2 and D3 receptors at energy levels of −9.8 and −9.4 kcal/mol, respectively. The DOM molecule interacted with the D2 receptor by forming a single hydrogen bond with the amino acid residue SER409 (1.89 Å). Furthermore, DOM formed six interactions with ASP114, PHE389, VAL91, LEU94, TYR408, and TRP413 amino acids. Figure 3 illustrates the 3D and 2D structures of the non-bond interactions between QUA and DOM with the D2 receptor. 

#### 2.2.2. QUA and OND with Serotoninergic (5HT3) Receptor Interaction

For the 5HT3 receptor, QUA exhibited a binding affinity of −7.9 kcal/mol, while the standard OND compound showed a binding affinity of −8.1 kcal/mol. QUA’s binding interaction with the 5HT3 receptor was facilitated by two hydrogen bonds with specific amino acid residues, namely ILE267 (2.51 Å) and ASP271 (2.95 Å), in addition to creating hydrophobic bonding interactions with ILE268, LEU266, LEU234, and PRO230 amino acids at the receptor binding site. On the other hand, OND’s binding interaction with the 5HT3 receptor was facilitated by two hydrophobic bonds with specific amino acid residues, namely LEU234 and VAL264 (Table 3). Figure 4 illustrates the 3D and 2D structures of the non-bond interactions between QUA and OND with the 5HT3 receptor.

#### 2.2.3. QUA and HYS with Muscarinic (M1, M2, M3, M4, and M5) Receptor Interaction

Both QUA and standard HYS exhibited higher affinity for the M4 receptor (Table 4). The binding affinities of QUA for the M1, M2, M3, M4, and M5 receptors were −7.8, −8.2, −9.0, −9.2, and −8.0 kcal/mol, respectively. The interaction between QUA and the M4 receptor involved the formation of two hydrogen bonds with specific amino acid residues, namely TYR416 (2.54 Å) and SER436 (2.06 Å). Additionally, QUA formed two hydrophobic bonds with PHE186 and TRP435 amino acid at the receptor binding site. In contrast, it is worth noting that HYS exhibited binding affinities with M1, M2, M3, M4, and M5 receptors at energy levels of −7.1, −7.5, −8.6, −8.9, and −7.7 kcal/mol, respectively. The HYS compound interacted with the M4 receptor by forming three hydrogen bonds with TYR92 (2.81 Å), ASP432 (2.71 Å), and ASN423 (3.70 Å) amino acids. Furthermore, HYS formed three hydrophobic bonding interactions with TYR439, PHE186, and TRP435 amino acids at the receptor binding site. Figure 5 illustrates the 3D and 2D structures of the non-bond interactions between QUA and HYS with the M4 receptor.

#### 2.2.4. In Silico Toxicity Predictions

The web server ProTox-II was utilized to predict toxicity parameters. In our in silico toxicity study, QUA exhibited a lethal dose 50 (LD50) of 159 mg/kg body weight with toxicity class 3 (Table 5). However, QUA showed no toxicity effects in terms of hepatotoxicity, immunotoxicity, and cytotoxicity. Nevertheless, QUA showed toxicity effects in terms of carcinogenicity and mutagenicity. In contrast, it has been observed that standard compounds, namely DOM and OND, exhibited immunotoxicity and mutagenicity toxicity, respectively. On the other hand, it has been determined that HYS does not demonstrate toxicity across all assessed parameters.

## 3. Discussion

Inadequate management of chemotherapy-induced nausea and vomiting (CINV) can significantly impact functional capabilities and may compromise treatment compliance [49]. The oral ingestion of toxic CuSO_4_·5H_2_O can lead to a distinct vagal emetic response due to its properties as an oxidizing agent and its corrosive effects on the gastrointestinal mucous membranes [50,51,52]. The induction of emesis occurs as a result of peripheral mechanisms, which involve the stimulation of visceral afferent nerve fibers in the GIT [53]. These stimuli are then transmitted to the vomiting center [54,55]. Recent findings demonstrated that 5HT3, D2, D3, H1, and M1–M5 receptors are present at the stimulation site and contribute to the induction of emesis. Antagonists of the 5HT3, D2, D3, H1, and M1-M5 receptors have the ability to prevent or significantly decrease emesis induced by chemotherapeutic agents [56,57].

The chosen standard drug, DOM, exhibited peripheral selectivity as an antagonist for D2 and D3 receptors [58]. Its mechanism of action involves the inhibition or antagonism of these receptors, which are located at the site of CTZ in the brain, thereby facilitating the desired therapeutic effects [59]. During our in vivo investigation, ingestion of the DOM group displayed a mean value of 11.80 ± 2.09 retches in chicks, while the mean value of retches in the NC group was 64.25 ± 3.69. The administration of OND and HYS decreased the frequency of retching episodes in the chick group compared to the control group receiving the vehicle. Based on empirical findings, it is possible to postulate that QUA demonstrated a safeguarding influence against toxicity by mitigating or preventing neural signals responsible for eliciting emetic responses. The significant reduction in the frequency of retches observed in both QUA groups compared to the NC group supports this notion, with the obtained mean values being 16.50 ± 4.65 and 10.00 ± 4.19 for the QUA-25 and QUA-50 groups, respectively. These values are close to or better than those of the standard groups. On the contrary, the different standard groups (DOM, OND, and HYS) administered to the animals demonstrated longer latency periods compared to the NC group. Specifically, the latency periods were 61.00 ± 3.34, 16.17 ± 3.64, and 11.50 ± 2.38 s for DOM, OND, and HYS, respectively. Interestingly, the treatment group (QUA-50) exhibited the highest latency period among all standard groups. The result shows that QUA-50 is more effective in reducing retches and prolonging latency periods compared to the standard DOM, OND, and HYS groups in CuSO_4_·5H_2_O-mediated emesis.

In the discipline of pharmacology, the phenomenon in which the combined impact of multiple medications surpasses the individual effects observed when each drug is administered in isolation is referred to as a synergistic effect or synergism [60]. Our experimental study demonstrated that using a combined drug therapy resulted in a reduction in the occurrence of retches and an elevation in the prolongation of the latency period in chicks, indicating the presence of a synergistic effect. Previous research has shown that administering antiemetic drugs effectively delays the occurrence of nausea or vomiting in response to emetic stimuli induced by cancer chemotherapy or acute toxicity [61]. Our study observed that the test combined group (DOM + QUA 50) displayed a significantly prolonged latency period of 156.67 ± 2.40 s compared to the NC group.

In studies conducted by Wang and Borison (1951) and Niijima et al. (1987), it was observed that CuSO_4_·5H_2_O does not exhibit the expected response to vagal nerve stimulation. Specifically, the researchers observed that vagotomy could not prevent emesis, a procedure that involves severing the distal portion of the vagus nerve in the GIT [62,63]. This suggests that the emetic response to CuSO_4_·5H_2_O may involve chemoreceptor signaling and gastrointestinal (GI). For this oral administration of QUA, the response protected the GI cells from damage and improved the vagal nervous system. Previous research showed that QUA has a GI protective effect against toxic drugs and bacterial infections [44,45]. Additionally, various studies revealed that QUA shows neuroprotective effects via maintaining the neurotransmitter, suppressing proinflammatory cytokines, and modulating the antioxidant signaling pathway [40]. Figure 6 illustrates a possible antiemetic mechanism of QUA and standard drugs.

Typically, the bioavailability of QUA is seen to be rather low and exhibits considerable inter-individual variability, despite a limited understanding of the underlying processes [64]. QUA, when administered in vivo to rats and pigs, has been seen to result in the presence of relatively low concentrations (ranging from picomolar to nanomolar levels) in brain tissue [65]. One crucial concern about the possible in vivo use of QUA is its ability to traverse the blood–brain barrier (BBB) [66]. Moreover, several in vitro investigations using BBB models consistently demonstrate the ability of QUA to penetrate the brain [67,68]. Specifically, the incorporation of QUA into lipid nanoparticles substantially enhances its ability to traverse the BBB and enter the brain [69,70]. Moreover, the study has shown that the simultaneous co-treatment of QUA and alpha-tocopherol leads to enhanced transportation of QUA over the BBB [71].

Molecular docking is a computational methodology employed to investigate the compatibility of a ligand with a receptor binding site, taking into consideration both geometric and energetic factors [72,73,74].The estimation of the interaction level between a ligand and a receptor is accomplished by assessing their binding affinity [75]. The binding interactions between QUA and the D2 receptor were higher compared to those of other receptors involved in the induction of emesis. The binding energy of QUA with D2 was −9.7 kcal/mol, whereas the docking value of standard DOM was −9.8 kcal/mol. The ligand–receptor interaction visualization indicates that the binding sites for QUA and DOM were SER193, TYR416, THR119, CYS118, ASP114, TRP386, and PHE390 for QUA, and SER409, ASP114, PHE389, VAL91, LEU94, TYR408, and TRP413 for DOM. In comparison, QUA binds different amino acids at the binding site of the D2 receptor and impedes the activation of the receptor. However, QUA shows only one binding similarity with the ASP114 amino acid. Multiple studies have consistently provided evidence indicating that the activation of D2 receptors triggers the vomiting center located in the CTZ [18,76,77]. QUA effectively inhibits the response of the D2 receptor, thereby impeding dopamine activity. Moreover, previous research revealed that QUA antagonized the action of the D2 receptor and improved neurogenic function [78]. Therefore, it can be confidently concluded that QUA exhibits strong inhibitory potency for the D2 receptor compared to other receptors responsible for emesis. This conclusion is supported by the higher docking scores of QUA with D2 receptors compared to other receptors. Additionally, in vivo, combined therapy with DOM + QUA-50 showed higher efficacy than other combinations.

Drug discovery and development is a protracted, financially burdensome, and precarious undertaking, spanning 10 to 15 years on average [79]. The approval of a novel pharmaceutical for clinical application entails an average expenditure exceeding USD 1 to USD 2 billion [80]. The drug candidate undergoes rigorous optimization during the preclinical stage before progressing to the phase I clinical trial [81]. Approximately 90% of drug candidates fail during the progression of clinical studies, specifically during phase I, II, and III clinical trials, primarily due to issues related to toxicity and a lack of clinical efficacy [82,83,84,85]. In silico toxicological investigations play a crucial and significant role in safe and cost-effective drug development [86]. In our present study, QUA demonstrated no toxic effects in terms of hepatotoxicity, immunotoxicity, and cytotoxicity. However, it did exhibit toxicity effects in terms of carcinogenicity and mutagenicity.

The study indicates that QUA exhibits significant antiemetic effects in response to CuSO_4_·5H_2_O-induced emesis, potentially attributed to its ability to antagonize D2 receptors. The in vivo findings also suggest that the antiemetic efficacy of QUA remains consistent and dependable when administered at a reduced dosage. 

Studies involving precise laboratory animals provide vital information about the benefits and drawbacks of novel drug candidates, as well as their potential biopharmaceutical implications [87,88]. Therefore, every pre-clinical investigation contributes to the assessment of biologically active compounds’ suitability for clinical trials by medical professionals. These investigations make it possible to determine the ideal routes of administration, the drug metabolic profile, the test dosage and frequency, and the creation of error-correction devices for use in clinical trials. This study has demonstrated that all conventional antiemetic medications distinctly suppress animals’ propensity towards emesis. The QUA in the tested sample also showed antiemetic activity in chicks in a dose-dependent manner. Compared to the standard and control groups, QUA demonstrated significant antiemetic effects in animals. Even though this prominent medication candidate has yet to be subjected to thorough toxicological investigations in animal models, in our study, QUA therapy did not produce any toxicological events, nor did it harm any chicks, demonstrating its reliability in this animal model. Additionally, our in vivo results support the findings of in silico research, suggesting that this bioactive may be taken into account in laboratory animal emesis. 

The main drawback of this study is that a number of factors might have an impact on the results, such as the differences in physiology and geometry of the animal’s stomach and the digestive capability of foods taken before the test started, which also impact emesis induction. Another possible limitation is the placement of the gavage tube at the time of inducer (copper sulfate) delivery in the gastric compartment; therefore, it is challenging to identify the precise chemical location and the emetic agent’s time of passage in the GI tract. In addition, this is a physiologic behavioral experiment; in this case, all animals would not respond similarly due to the variance of the environment, such as laboratory light, noise during the experiment, and test time differences, although we followed the optimum laboratory protocols mentioned in the study design section.

## 4. Materials and Methods

### 4.1. Chemical Reagents and Standards

The compound Quercetin (QUA) [2-(3,4-dihydroxyphenyl)-3,5,7-trihydroxychromen-4-one], with a purity of 98%, extra pure grade yellow color, and a CAS number of 6151-25-3, was acquired from Sisco Research Laboratories Pvt. Ltd. (Mumbai, India). Copper sulfate pentahydrate (CuSO_4_·5H_2_O) and Tween 80 were obtained from Merck (Mumbai, India). The reference/standard drugs Ondansetron (OND), Hyoscine butyl bromide (HYS), and Domperidone (DOM) were obtained from Incepta Pharma. Ltd. (Dhaka, Bangladesh), Opsonin Pharma. Ltd. (Barishal, Bangladesh), and Beximco Pharma. Ltd. (Dhaka, Bangladesh), respectively, all located in Bangladesh.

### 4.2. Preparation of Test and Standard Drugs

We selected two (lower and higher) doses of the test compound (QUA) based on a literature review. We prepared the parent solution of the test compound at a 50 µg/mL concentration by adding a small amount of Tween 80 as a co-solvent and then adjusted it with distilled water (DW). After that, the parent solution was diluted to achieve a concentration of 25 µg/mL. Additionally, the solutions of standard drugs (DOM, OND, and HYS) were also prepared by thoroughly mixing them into DW after adding a small amount of Tween 80 at concentrations of 6, 5, and 21 µg/mL for DOM, OND, and HYS, respectively. The higher dose (50 mg/kg) of QUA was chosen for administration in the combination treatment because it exhibited the most favorable results when treated alone to understand the combination effect more precisely. 

### 4.3. Animals

Young chickens (*Gallus gallus domesticus*) of both genders, aged 3 days, with an average weight ranging from 40 to 48 gm (Grade-A), were obtained from Nourish Poultry & Hatchery Ltd., located on Sonargaon Janapath Road, Uttara, Dhaka-1230, Bangladesh. The chickens were housed in groups in stainless steel enclosures that featured an upper hood opening. The enclosures were maintained at room temperature, subjected to a dark cycle, and kept in the presence of light for a duration of twelve hours. The individuals were granted unrestricted access to regular provisions of food and water. An additional day was allocated for preparatory activities prior to the commencement of the experiment, following the completion of data collection from suppliers. Before the antiemetic test, chicks were starved for about 12 h. This study was approved and funded by the Bangabandhu Sheikh Mujibur Rahman Science and Technology University Research Center (BSMRSTU-RC) (Approval No. 2023-33). 

### 4.4. In Vivo Investigation

#### 4.4.1. CuSO_4_·5H_2_O-Induced Emesis Model in Broiler Chicks

The investigation was carried out with slight modifications to the methodologies outlined by [89]. All the chicks were allocated by dividing them into nine groups, each consisting of five individuals. Before receiving the treatments, each chick specimen was housed in a spacious, transparent receptacle made of plastic for 10 min. Two different doses (25 and 50 mg/kg) of the QUA test sample were given orally based on information obtained from a literature review. The positive control (PC) drugs, namely DOM, OND, and HYS, were administered orally at doses of 6, 5, and 21 mg/kg body weight (b.w.), respectively. Three doses of the standard drugs were prepared by combining them with QUA at 50 mg/kg. These combined doses were then ingested orally by animals to assess any potential synergistic effects. This study designated the compound Tween 80 and DW mixture as the negative control (NC). It was administered orally at 150 mg/kg b.w. Following a treatment duration of 30 min, CuSO_4_·5H_2_O was given orally to each chick specimen at a dose of 50 mg/kg of b.w. in order to initiate emesis. Subsequently, the latency period, defined as the time interval between the administration of CuSO_4_·5H_2_O treatment and the occurrence of the first retch, as well as the total number of retches within a 10 min timeframe following the administration of CuSO_4_·5H_2_O treatment, was meticulously documented. A blank control (without treatment) was also taken for this study. The equations used to calculate the percentage increase in latency and decrease in retches inspection of NC were as follows:$$\%increase\;in\;latency=\frac{B-C}{B}\times 100$$
$$\%decrease\;in\;retches=\frac{X-Y}{X}\times 100$$
where B = mean of latency in seconds in standard and test groups, C = mean of latency in seconds in NC group, X = mean of retches in NC group, Y = mean of retches in standard and test groups.

#### 4.4.2. Statistical Analysis

The antiemetic activity values are presented as the mean value, accompanied by the standard error of the mean (SEM). The difference’s statistical significance was calculated using Graph Pad Prism (version 9.0), a statistical software program. A 95% confidence interval was employed for this determination. Significance was attributed to *p*-values less than 0.05, while *p*-values less than 0.0001 were deemed highly significant.

### 4.5. In Silico Investigation

#### 4.5.1. Receptor Selection and Preparation

The literature review identified eight receptors that were implicated in the induction of emesis. The three-dimensional (3D) structures of the targeted receptors, namely 5HT3 (PDB ID: 6Y5B), D2 (PDB ID: 6CM4), D3 (PDB ID: 3PBL), M1 (PDB ID: 6WJC), M2 (PDB ID: 5ZK8), M3 (PDB ID: 4U15), M4 (PDB ID: 7V6A), and M5 (PDB ID: 6OL9), were obtained from the Protein Data Bank (https://www.rcsb.org/, accessed on 25 September 2023). Following the collection process, the receptors were subjected to optimization procedures aimed at mitigating potential docking interference. This was carried out by eliminating extraneous molecules, such as water molecules, lipids, and heteroatoms, from the protein sequence using the PyMol software package (version 2.4.1) [90]. The Swiss-PDB Viewer software (version 4.1.0) program was used to optimize the receptors’ design and minimize energy consumption, employing the GROMOS96 force field [91]. As a result, a Protein Data Bank (PDB) file was created and stored for a subsequent molecular docking investigation.

#### 4.5.2. Ligand Collection and Preparation

The 3D conformers of DOM (Compound CID: 3151), OND (Compound CID: 4595), HYS (Compound CID: 3000322), and QUA (Compound CID: 5280343) were obtained in the structure-data file (SDF) format of a compound from the PubChem database (https://pubchem.ncbi.nlm.nih.gov/, accessed on 25 September 2023). Subsequently, the 3D conformers of the chemical compounds were optimized to minimize internal energies using the Chem3D 16.0 program package, which is commonly employed for molecular docking studies [92,93]. Figure 7 exhibits the visual representations of the chemical agents in a two-dimensional (2D) format.

#### 4.5.3. Molecular Docking Protocol

Molecular docking is a computational tool employed in the field of medicinal chemistry for the purpose of drug design. The active binding affinity of the ligands against the receptors’ active areas was estimated through the utilization of the PyRx software (version 0.8) package in the process of molecular docking. To facilitate the docking process, the dimensions of the grid box were adjusted to their maximum values along the x-, y-, and z-axes. The calculation was then executed over a span of 2000 steps [94,95]. The docking potential result was saved in files in the comma-separated values (‘csv’) format, while the ligand–receptor complex was obtained in PDB format in order to collect the ligand in PDBQT format. After that, the ligand–receptor interactions and their corresponding active sites were examined using the PyMol (v2.4.1) and Discovery Studio Visualizer (v21.1.020298) software packages. Subsequently, bond types, the number of hydrogen bonds (HBs), the amino acid residues, the length of HBs, and other bond types for each ligand–receptor interaction were documented [96,97].

#### 4.5.4. Toxicity Prediction

The ProTox-II web servers were employed to predict toxicity parameters. The objective of the ProTox II web server is to analyze a range of parameters associated with hepatotoxicity, carcinogenicity, immunotoxicity, mutagenicity, and cytotoxicity [98,99]. The chemical agent’s data obtained from PubChem, including SMILES (Simplified Molecular Input Line-Entry System) representations, were inputted into the search bars of ProTox II for analysis.

## 5. Conclusions

The findings of this study provide evidence that QUA exhibits notable antiemetic properties and effectively mitigates CuSO_4_·5H_2_O-induced retching in chicks, potentially through its peripheral mechanism of action. The outcomes of the molecular docking analysis indicate that QUA is more able to bind to dopamine receptors, particularly D2 receptors, compared to other receptors known to induce emesis. Our test compound, QUA, also shows synergistic effects when administered in conjunction with established antiemetic drugs. Therefore, the compound QUA might be a good option in the treatment of emesis. Similarly, the computational toxicity (random forest (RF) algorithm method) analysis reveals that QUA demonstrates toxic properties in relation to carcinogenicity and mutagenicity. Further animals and more desirable studies (human organs on chips) are required to check the validity of the toxicity of QUA. Additionally, it is unclear how QUA exactly impedes the emesis, since the molecular pathways and which receptors that are potential targets for emesis may not have been fully understood. The implications of these findings extend to future non-clinical, preclinical, and clinical investigations, necessitating further in vivo studies to validate the observed outcomes. Researchers in the field of medicinal chemistry must be encouraged to conduct thorough investigations into this highly promising natural lead compound and its various derivatives.

## Figures and Tables

**Figure 1 plants-12-04189-f001:**
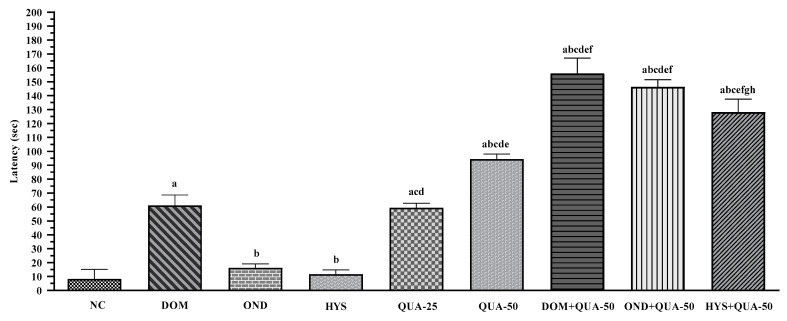
Latency (second) of retches observed in the test sample, controls, and combinations (NC: Negative control; DOM: Domperidone (6 mg/kg); OND: Ondansetron (5 mg/kg); HYS: Hyoscine (21 mg/kg); QUA: Quercetin (25 and 50 mg/kg). Values are mean ± S.E.M. (*n* = 5). ^a^ compared to the NC (vehicle), ^b^ compared to the DOM; ^c^ compared to the OND; ^d^ compared to the HYS; ^e^ compared to the QUA-25; ^f^ compared to the QUA-50; ^g^ compared to the DOM + QUA-50; ^h^ compared to the OND + QUA-50; *p* < 0.05 (OND + QUA-50 vs. HYS + QUA-50); *p* < 0.0001(NC vs. DOM, NC vs. QUA-25, NC vs. QUA-50, NC vs. DOM + QUA-50, NC vs. OND + QUA-50, NC vs. HYS + QUA-50, DOM vs. OND, DOM vs. HYS, DOM vs. QUA-50, DOM vs. DOM + QUA-50, DOM vs. OND + QUA-50, DOM vs. HYS + QUA-50, OND vs. QUA-25, OND vs. QUA-50, OND vs. DOM + QUA-50, OND vs. OND + QUA-50, OND vs. HYS + QUA-50, HYS vs. QUA-25, HYS vs. QUA-50, HYS vs. DOM + QUA-50, HYS vs. OND + QUA-50, HYS vs. HYS + QUA-50, QUA-25 vs. QUA-50, QUA-25 vs. DOM + QUA-50, QUA-25 vs. OND + QUA-50, QUA-25 vs. HYS + QUA-50, QUA-50 vs. DOM + QUA-50, QUA-50 vs. OND + QUA-50, QUA-50 vs. HYS + QUA-50, DOM + QUA-50 vs. HYS + QUA-50)).

**Figure 2 plants-12-04189-f002:**
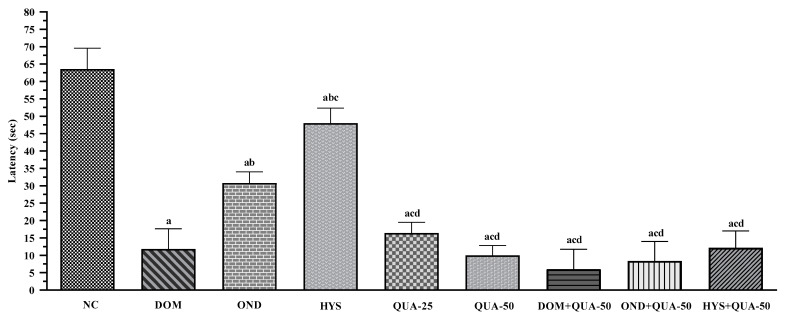
Number of retches observed in the test sample, controls, and combinations (NC: Negative control; DOM: Domperidone (6 mg/kg); OND: Ondansetron (5 mg/kg); HYS: Hyoscine (21 mg/kg); QUA: Quercetin (25 and 50 mg/kg). Values are mean ± S.E.M. (n = 5). ^a^ compared to the NC (vehicle), ^b^ compared to the DOM; ^c^ compared to the OND; ^d^ compared to the HYS; *p* < 0.05 (NC vs. HYS, OND vs. QUA-25); *p* < 0.01 (DOM vs. OND, OND vs. HYS, OND vs. HYS + QUA-50); *p* < 0.001 (OND vs. QUA-50, OND vs. OND + QUA-50); *p* < 0.0001 (NC vs. DOM, NC vs. OND, NC vs. QUA-25, NC vs. QUA-50, NC vs. DOM + QUA-50, NC vs. OND + QUA-50, NC vs. HYS + QUA-50, DOM vs. HYS, OND vs. DOM + QUA-50, HYS vs. QUA-25, HYS vs. QUA-50, HYS vs. DOM + QUA-50, HYS vs. OND + QUA-50, HYS vs. HYS + QUA-50)).

**Figure 3 plants-12-04189-f003:**
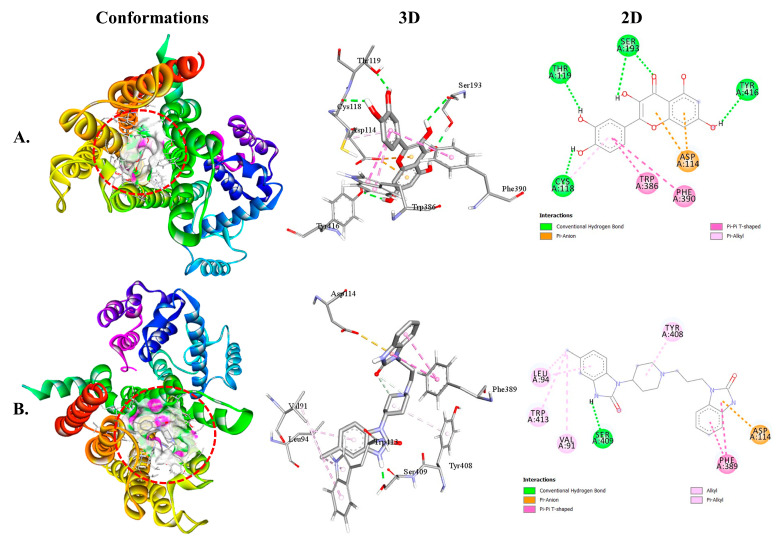
The best possible interaction for molecular docking of the D2 receptor with (**A**) QUA or (**B**) DOM compounds.

**Figure 4 plants-12-04189-f004:**
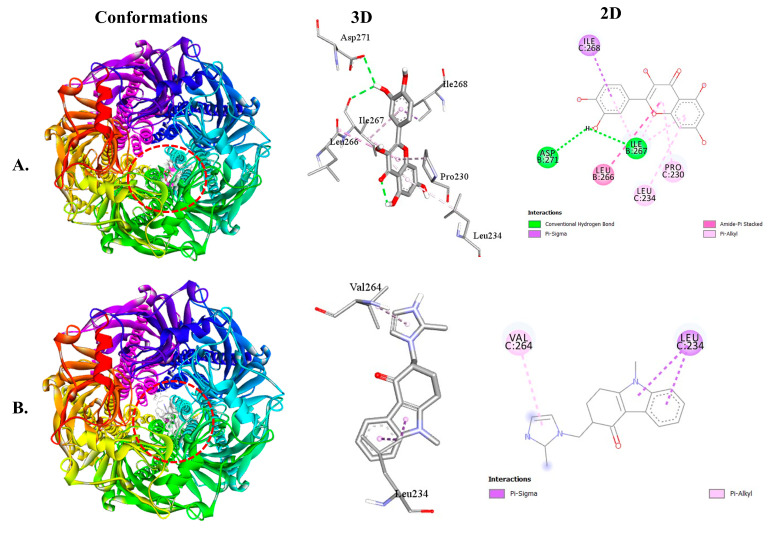
The best possible interaction for molecular docking of the 5HT3 receptor with (**A**) QUA or (**B**) OND compounds.

**Figure 5 plants-12-04189-f005:**
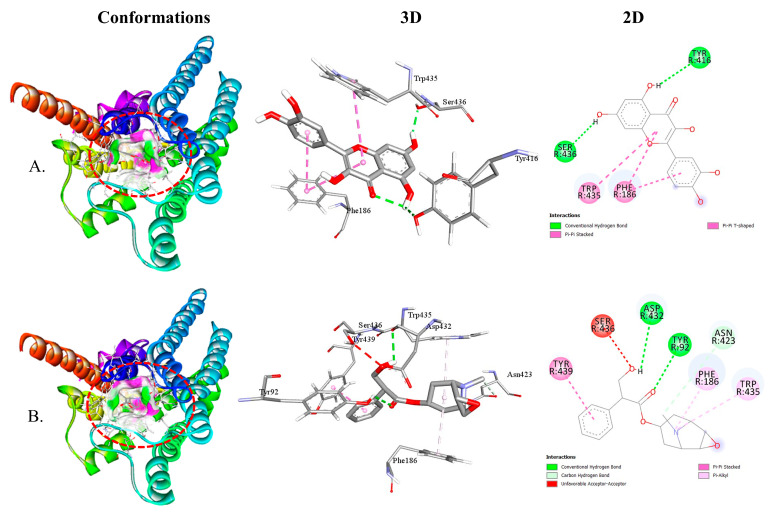
The best possible interaction for molecular docking of the M4 receptor with (**A**) QUA or (**B**) HYS compounds.

**Figure 6 plants-12-04189-f006:**
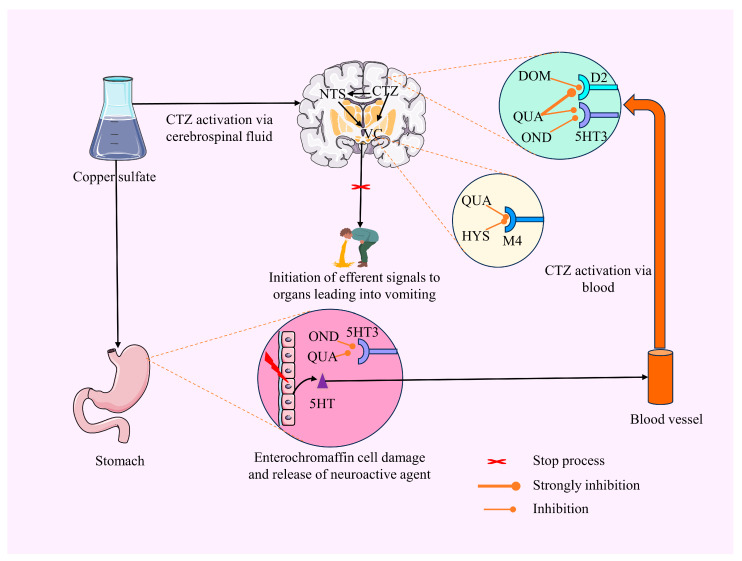
The proposed mechanism of action for the antiemetic effects (QUA: Quercetin; DOM: Domperidone; OND: Ondansetron; HYS: Hyoscine; CTZ: Chemoreceptor Trigger zone; NTS: Nucleus Tractus Solitaries; VC: Vomiting center; 5HT: 5 Hydroxy tryptamines; 5HT3: Serotonin receptor 3; D2: Dopamine receptor 2; M4: Muscarinic receptor 4) of QUA, DOM, OND, and HYS is based on their binding affinity with the D2, 5HT3, and M4 receptors. (Briefly, QUA is evidently bound to 5HT3) [46]. Our in silico studies suggest that it can also bind with D2 and M4 receptors. Upon reviewing the literature report and our present study, we suppose QUA may inhibit these three receptors. On the other hand, both DOM and OND display inhibitory effects on D2 and 5HT3 receptors, respectively, while HYS specifically blocks the M4 receptor. By antagonizing these stomach receptors, these drugs prevent the formation of the vomiting center located in the medulla oblongata in animals. As a result, there is a lack of GIT contraction, muscle contraction, and the initiation of efferent signals to organs, ultimately inhibiting emesis).

**Figure 7 plants-12-04189-f007:**
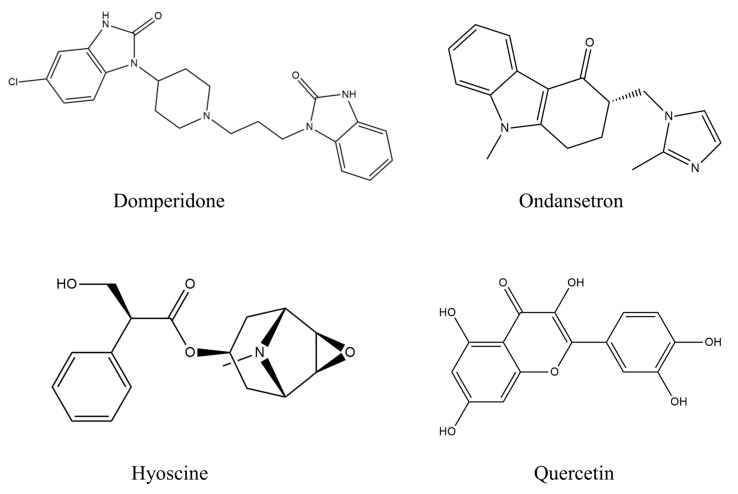
Structures of Quercetin and the selected standard were screened for the emesis-inducing receptor.

**Table 1 plants-12-04189-t001:** The treatment groups’ percentage increase in latency and percentage reduction in retches observed in the test and referral groups.

Name of Group	%Increase in Latency	%Decrease in Retches
NC (vehicle)	-	-
DOM	86.48	81.63
OND	48.98	52.06
HYS	28.26	24.00
QUA-50	91.25	84.43
QUA-25	86.08	74.32
DOM + QUA-50	94.73	90.66
OND + QUA-50	94.36	87.04
HYS + QUA-50	93.56	81.01

NC: Distilled water (Dose: 150 mg/kg); DOM: Domperidone (Dose: 6 mg/kg); OND: Ondansetron (Dose: 5 mg/kg); HYS: Hyoscine (Dose: 21 mg/kg); QUA-50: Quercetin (Dose: 50 mg/kg); QUA-25: Quercetin (Dose: 25 mg/kg); DOM + QUA-50: Domperidone + Quercetin (Dose: 6 + 50 mg/kg); ODN + QUA-50: Ondansetron + Quercetin (Dose: 5 + 50 mg/kg); HYS + QUA-50: Hyoscine + Quercetin (21 + 50 mg/kg).

**Table 2 plants-12-04189-t002:** The best results of a molecular docking study of QUA and DOM with two (D2 and D3) receptors.

Receptors	Binding Affinity	No. of HB	HB Residues	HB Length (Å)	Other Bond Residues
D2-QUA	−9.7	4	SER193 TYR416 THR119 CYS118	1.85 2.20 2.14 2.18	ASP114 TRP386 PHE390
D3-QUA	−8.5	3	VAL189 TYR365 VAL111	2.36 2.09 3.56	ASP110 ILE183 PHE345 HIS349 VAL107 ILE183
D2-DOM	−9.8	1	SER409	1.89	ASP114 PHE389 VAL91 LEU94 TYR408 TRP413
D3-DOM	−9.4	3	GLY1107 VAL1103 PHE1104	2.83 1.77 3.53	GLU1011 ASP1020 LEU1032 ALA1074

QUA: Quercetin; DOM: Domperidone; HB: Hydrogen bond.

**Table 3 plants-12-04189-t003:** The best results of a molecular docking study of QUA and OND with 5HT3 receptors.

Receptors	Binding Affinity	No. of HB	HB Residues	HB Length (Å)	Other Bond Residues
5HT3-QUA	−7.9	2	ILE267 ASP271	2.51 2.95	ILE268 LEU266 ILE267 LEU234 PRO230
5HT3-OND	−8.1	-	-	-	LEU234 VAL264

QUA: Quercetin; OND: Ondansetron; HB: Hydrogen bond.

**Table 4 plants-12-04189-t004:** The best results of a molecular docking study of QUA and HYS with five (M1, M2, M3, M4, and M5) receptors.

Receptors	Binding Affinity	No. of HB	HB Residues	HB Length (Å)	Other Bond Residues
M1-QUA	−7.8	2	TYR106 ILE180	2.03 2.77	TYR404
M2-QUA	−8.2	2	PHE181 TYR104	2.43 1.96	TRP422
M3-QUA	−9	2	TRP206 ARG183	2.08 3.22	ARG171 MET187 VAL210 TYR175
M4-QUA	−9.2	2	TYR416 SER436	2.54 2.06	PHE186 TRP435
M5-QUA	−8	2	ILE185 HIS478	2.76 2.19	TYR481 TRP477
M1-HYS	−7.1	4	THR189 CYS178 TYR404 LEU183	2.01 2.33 3.62 3.41	TYR179
M2-HYS	−7.5	-	-	-	TYR426
M3-HYS	−8.6	2	ASN507 TYR529	2.26 3.47	CYS532 TYR148 TRP503 TYR506 ALA235
M4-HYS	−8.9	3	TYR92 ASP432 ASN423	2.81 2.71 3.70	TYR439 PHE186 TRP435
M5-HYS	−7.7		ILE185 HIS478 SER189	1.93 2.62 2.95	TRP477 TYR481

QUA: Quercetin; HYS: Hyoscine; HB: Hydrogen bond.

**Table 5 plants-12-04189-t005:** The toxicity prediction of four compounds, namely QUA, DOM, OND, and HYS, using the ProTox-II model.

Properties	Parameters	QUA	DOM	OND	HYS
Toxicity	LD50	159 mg/kg	715 mg/kg	95 mg/kg	1275 mg/kg
	Toxicity class	3	4	3	4
	Hepatotoxicity	Inactive	Inactive	Inactive	Inactive
	Carcinogenicity	Active	Inactive	Inactive	Inactive
	Immunotoxicity	Inactive	Active	Inactive	Inactive
	Mutagenicity	Active	Inactive	Active	Inactive
	Cytotoxicity	Inactive	Inactive	Inactive	Inactive

QUA: Quercetin; DOM: Domperidone; OND: Ondansetron; HYS: Hyoscine; LD50: Lethal dose 50.

## Data Availability

Data are contained within this article.

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
