# Peer review of "Assessment of Quercetin Antiemetic Properties: In Vivo and In Silico Investigations on Receptor Binding Affinity and Synergistic Effects"

_plants, 2023, doi:10.3390/plants12244189_

Round 1
Reviewer 1 Report
Comments and Suggestions for Authors
The manuscript is covering the antiemetic effect of quercetin by using animal test and an in silico study. It is interesting to find that the oral adminstration of two doses of quercetin effectively prevents the emesis induced by copper sulfate. Quercetin's antiemetic effect surpasses that of three antiemetic drugs: ondansetron (OND), domperidone (DOM), and hyoscine butyl bromide (HYS). Moreover, combining quercetin with OND and DOM shows significantly stronger antiemetic effects. Additionally, quercetin exhibits a high binding affinity with various emesis-related receptors, including D2, D3, 5HT3, M1, M2, M3, M4, and M5 receptors. These findings offer new insights into the health benefits of quercetin.
Remarks:
1. Quercetin's limited bioavailability results in a small portion reaching the blood after oral intake. Whether intact quercetin can cross the blood-brain barrier to reach multiple brain receptors is questionable. Authors should consider these factors when discussing the mechanism of quercetin's antiemetic effect.
2. The in silico study results need stronger justification through experimental data. Detecting specific downstream signal molecules responding to D2 receptor activation is suggested.
3. Include a blank control in animal testing, without copper sulfate treatment.
4. Shorten the introduction, focusing more on quercetin's antiemetic or related pharmacological effects.
5. Strengthen the discussion by avoiding repetitive results, reflecting more on the obtained data, and comparing it with existing references.
Author Response
The manuscript is covering the antiemetic effect of quercetin by using animal test and an in silico study. It is interesting to find that the oral adminstration of two doses of quercetin effectively prevents the emesis induced by copper sulfate. Quercetin's antiemetic effect surpasses that of three antiemetic drugs: ondansetron (OND), domperidone (DOM), and hyoscine butyl bromide (HYS). Moreover, combining quercetin with OND and DOM shows significantly stronger antiemetic effects. Additionally, quercetin exhibits a high binding affinity with various emesis-related receptors, including D2, D3, 5HT3, M1, M2, M3, M4, and M5 receptors. These findings offer new insights into the health benefits of quercetin.
Authors: Thank you for your efforts and suggestions to develop the manuscript. We think your comments and suggestions will make our manuscript better and easier to understand for readers.
Remarks:
- Quercetin's limited bioavailability results in a small portion reaching the blood after oral intake. Whether intact quercetin can cross the blood-brain barrier to reach multiple brain receptors is questionable. Authors should consider these factors when discussing the mechanism of quercetin's antiemetic effect.
Authors: We would like to thank you for your comment. We discussed the quercetin low bioavailability results in the quercetin antiemetic mechanisms in the discussion section of our updated manuscript.
- The in silico study results need stronger justification through experimental data. Detecting specific downstream signal molecules responding to D2 receptor activation is suggested.
Authors: We added the informative sentences for the in silico study for stronger justification through experimental data.
- Include a blank control in animal testing, without copper sulfate treatment.
Authors: We did it. As this group did not produce any emesis or emetic tendencies, we did not show it in the main figure. We have added relevant lines in the results and methods sections.
- Shorten the introduction, focusing more on quercetin's antiemetic or related pharmacological effects.
Authors: We have reduced and added relevant information on quercetin's antiemetic or related pharmacological effects to make the introduction more precise. We mainly added the gastroprotective data and antiemetic mechanisms to our revised manuscript.
- Strengthen the discussion by avoiding repetitive results, reflecting more on the obtained data, and comparing it with existing references.
Authors: Thank you for your valuable suggestion. We have added relevant information in the discussion section according to your suggestions, and we have interpreted the results with existing literature; therefore, we have kept the data from our investigation in the discussion section to make it more understandable to the readers.
Reviewer 2 Report
Comments and Suggestions for Authors
The authors studied the antiemetic activity of quercetin using in silico and in vivo models. The manuscript is well organized and data presented in this work are interesting for publication. I will recommend its publication after major revisions suggested below:
Title: Keep each word capitalized as per the target journal requirements.
Keywords: Kindly arrange all keywords in alphabetical manner.
Introduction: Kindly mention the reported biological activities of quercetin. Also include the novelty of your study.
Figures 2-6: Please improve the resolution and quality of these figures.
The text in each headings and sub-headings must contain each word capitalized.
Results and discussion: Please compare your results with previous studies and mention clearly how your work is important in comparison to already been reported.
Authors are advised to include the main limitation of work at the end of results and discussion section and just before the conclusion.
Avoid abbreviations before giving their explanation in the abstract, text, figure, and table. After the definition of each abbreviation, no need to write its full form.
Conclusion: The conclusion should be concise and to the point indicating the application of the work. What are future directions of the present investigation? Kindly include them in the conclusion section.
Acknowledgement: The authors acknowledged “Researchers Supporting Project number (RSPD2023R744), King Saud University, Riyadh, Saudi Arabia”. However, there is no author/collaborator from King Saud University in the author list. Could authors explain how did RSPD Unit at King Saud University provided this support?
Comments on the Quality of English Language
Minor editing of the language is required.
Author Response
The authors studied the antiemetic activity of quercetin using in silico and in vivo models. The manuscript is well organized and data presented in this work are interesting for publication. I will recommend its publication after major revisions suggested below:
Authors: We would like to express our cordial gratitude to you for your appreciation and recommendation regarding our manuscript.
Title: Keep each word capitalized as per the target journal requirements.
Authors: We have capitalized the first letter of each word of the title as per the journal requirements in the revised manuscript.
Keywords: Kindly arrange all keywords in alphabetical manner.
Authors: Done
Introduction: Kindly mention the reported biological activities of quercetin. Also include the novelty of your study.
Authors: Dear reviewer, we have clearly mentioned the biological activity of quercetin related to gatroprotective activities in the introduction section, as well as the novelty of our study in discussion and conclusion sections. Quercetin, a flavonoid compound, has a wide range of biological applications. However, we have not found any antiemetic studies of quercetin, which is why we designed our study to evaluate the antiemetic activity of quercetin in a chick model, and this study provided evidence of the antiemetic effect and its related mechanism for the first time.
Figures 2-6: Please improve the resolution and quality of these figures.
Authors: Thank you for your comment. We have enhanced the quality of Figure 2-6 and incorporated a high-resolution image into the revised manuscript.
The text in each heading and sub-headings must contain each word capitalized.
Authors: Done
Results and discussion: Please compare your results with previous studies and mention clearly how your work is important in comparison to already been reported.
Authors: We have added relevant information to the discussion section according to your suggestions. We have not found any reports of quercetin having an antiemetic effect. However, there are several previous reports on gastroprotective effects, and we have interpreted the present data with them to make the application of GIT disturbances more scientific.
Authors are advised to include the main limitation of work at the end of results and discussion section and just before the conclusion.
Authors: Thank you for your valuable comment. We added the major limitations of our study in the last paragraph of the discussion section before the methodology section.
Avoid abbreviations before giving their explanation in the abstract, text, figure, and table. After the definition of each abbreviation, no need to write its full form.
Authors: Done
Conclusion: The conclusion should be concise and to the point indicating the application of the work. What are future directions of the present investigation? Kindly include them in the conclusion section.
Authors: We agree with you; we have improved the conclusion section and added the future direction of the present study to our revised manuscript.
Acknowledgement: The authors acknowledged “Researchers Supporting Project number (RSPD2023R744), King Saud University, Riyadh, Saudi Arabia”. However, there is no author/collaborator from King Saud University in the author list. Could authors explain how did RSPD Unit at King Saud University provided this support?
Authors: It was a mistake. We have corrected the point.
Comments on the Quality of English Language
Minor editing of the language is required.
Authors: Done
Reviewer 3 Report
Comments and Suggestions for Authors
In the article, the authors present experimental results regarding the potential antiemetic properties of quercetin using the concept of in vivo and in silico.
Based on experimental evidence, they proposed reaction mechanisms that are OK with me.
Please clarify two issues:
1. on what basis was the dose of quercetin 25 and 50 mg/kg used in the experiment developed?
2. The authors state that quercetin was dissolved in distilled water and administered orally to the colts based on information obtained from the literature review (line 149-151). The reviewer's question: how was quercetin dissolved since it is practically insoluble in water? What literature data do the authors refer to?
Author Response
In the article, the authors present experimental results regarding the potential antiemetic properties of quercetin using the concept of in vivo and in silico.
Based on experimental evidence, they proposed reaction mechanisms that are OK with me.
Authors: Dear reviewer, Thank you for your valuable observation and comment. We are grateful for your positive feedback on our proposed reaction mechanisms.
Please clarify two issues:
- on what basis was the dose of quercetin 25 and 50 mg/kg used in the experiment developed?
Authors: We selected the investigational doses and combinational doses on the basis of existing literatures. ref:
- https://doi.org/10.1248/bpb.26.1398
- https://doi.org/10.1016/j.ejphar.2011.08.014
- https://doi.org/10.1080/10715760310001616014
- The authors state that quercetin was dissolved in distilled water and administered orally to the colts based on information obtained from the literature review (line 149-151). The reviewer's question: how was quercetin dissolved since it is practically insoluble in water? What literature data do the authors refer to?
Authors: Thank you for your valuable comment. We prepared the solution of quercetin by adding a small amount of tween 80 directly to the quercetin powder (which acts as a co-solvent) and then added the required amount of DW to confirm the concentration used. However, we have corrected the methods and materials section by adding a solution preparation procedure to avoid any conflict.
Round 2
Reviewer 1 Report
Comments and Suggestions for Authors
The authors well addressed my comments.
Reviewer 2 Report
Comments and Suggestions for Authors
The authors have addressed the previous concerns. Revised manuscript is suitable for publication in its present form.
Reviewer 3 Report
Comments and Suggestions for Authors
Thank you for making the corrections.